# OpenReview forum: "Towards Non-destructive Privacy Protection for LVLMs via node-level localized editing"
_ICLR.cc/2026/Conference — ICLR 2026 Conference Withdrawn Submission_

### Official Review · Reviewer_uawP · 2025-10-27

**Soundness:** 1
**Presentation:** 1
**Contribution:** 2
**Rating:** 2
**Confidence:** 4

**Summary:**

This paper aims to mitigate the privacy leakage problem in large vision-language models (LVLMs). The authors propose PRN-Edit, a two-stage localised editing method designed to modify the models’ responses to predefined privacy-sensitive queries.

**Strengths:**

- **Novel Problem Definition** The paper proposes to use a distinct editing problem to address the privacy risk, which seems to be practical and relevant for real-world deployment of LVLMs

- **Methodological Design and Motivation**  The proposed neuron-level localisation and editing pipeline presents a technically interesting approach to fine-grained model intervention, offering a reasonable degree of innovation, particularly in aligning privacy mitigation to preserve model generality. The emphasis on non-destructive editing—maintaining task performance while suppressing privacy sensitive behaviours is well motivated and supported by empirical results.

- **Evaluation Design and Analysis** The authors conduct extensive experiments by constructing a paired privacy dataset and further evaluating their approach against recent model-editing baselines on the ScienceQA and MLLMGuard benchmarks.

**Weaknesses:**

1. **Scope Alignment and Relevance to LVLMs** The proposed technique does not appear to be tightly coupled with the LVLM-specific problem setting described in the paper. While the method is evaluated on multimodality models, its core mechanism, localised parameter editing, could in principle be applied to any LLMs and LVLMs. Moreover, most of the compared baselines (e.g., MEMIT, DINM) were originally designed for text-only LLMs, which raises concerns about whether the claimed contributions are genuinely tailored to the multimodality privacy challenge.

2. **Clarity, Generalisation, and Sensitivity Issues** Certain parts of the current version lack conceptual clarity and suffer from an overly complex narrative. The proposed framework seems to rely heavily on a specific definition of privacy-sensitive questions and requires two additional optimisation stages. As shown in Table 5, the method is quite sensitive to hyperparameters and the choice of the edited network layer(s). These factors collectively cast doubt on the robustness and generalisability of the approach beyond the tested settings.

3. **Overlap with LLM Guardrails and Missing Discussion** The problem addressed in this paper overlaps conceptually with LLM safety guardrails, which both aim to enforce safe or privacy-preserving model behaviour. However, the paper does not provide a clear discussion or experimental comparison with guardrail-based approaches. Including such analysis would help clarify whether PRN-Edit offers complementary advantages or merely reimplements an internalised version of existing behavioural defences.

**Questions:**

1. Could the authors provide more detailed explanations of the loss functions defined in Equations (3) and (4)?

2. On page 5, lines 262–267, referring to $M_l$ as a mask seems somewhat inaccurate, since it functions more like an additional linear layer. Moreover, the relationship between the elements of $M_l$ and the cosine-based interpretation is not clearly justified. Simply stating that negative elements correspond to privacy-risky nodes makes more sense.

---

### Official Review · Reviewer_hdga · 2025-10-28

**Soundness:** 2
**Presentation:** 1
**Contribution:** 2
**Rating:** 2
**Confidence:** 3

**Summary:**

The paper presents an important problem of safeguarding the large vision language models that struggle to consistently refuse the privacy-compromising instructions from the user. The paper proposes PRN-Edit, which is a privacy risk mitigation method based on model editing. It improves the model's privacy protection by increasing the rate of refusal to answer privacy-related questions. It can also generalize well to unseen samples. It uses a learnable feature mask to locate the privacy risk nodes in the feature encoding of the user instructions, which then precisely guides the update of the model parameters. Experimental results on LVLMs such as MiniGPT-4 and LLava-1.5 further verify the claims.

**Strengths:**

1. The problem taken up is important and useful for the community (though not being solved for the first time).

2. The major advantage is that the method generalizes well to unseen samples, which is the outcome of the better dataset creation for training the model.

**Weaknesses:**

1. The paper is not very well written, and there are multiple missing details (which are there in appendix), making the solution confusing; for instance, the authors did not even explain what $\mathcal{L}_{sen}$ looks like in the main text (which is the main contribution) and deferring it to the Appendix, and there is no information on what is frozen and what is being learned during the module 2 training. There is no detailed description of why the values of $\alpha$ and $\beta$ are chosen as such.

2. Although the problem is being phrased as LVLM privacy protection, the underlying solution only works in the LLM part, where the input query is being observed and decisions are made based on that. For instance, the example given in the paper, that given a passport attacker can ask for the passport number, hence the problem is only in the language part, as per my understanding. This makes all the existing privacy protection methods candidates as baselines.

3. The paper does not discuss anything about why the existing LLM privacy protection methods cannot be used. As the major processing is performed on the LLM part of the backbone why the existing LLM-based methods for privacy protection are not applicable, and are not being considered as a baseline.

**Questions:**

See Weaknesses.

---

### Official Review · Reviewer_V8rL · 2025-10-30

**Soundness:** 3
**Presentation:** 2
**Contribution:** 3
**Rating:** 4
**Confidence:** 3

**Summary:**

The paper addresses privacy risks in Large Vision-Language Models, focusing on scenarios where models may follow sensitive user instructions, potentially exposing private information even if it was not in the training data. To mitigate this, the authors propose PRN-Edit, a localized feature-level model editing method that identifies privacy-risk nodes in the feed-forward layers of Transformers and selectively updates them to increase refusal rates for sensitive queries. They also construct a paired-sample dataset covering six privacy categories to help models distinguish between sensitive and benign questions. Experiments on MiniGPT-4 and LLaVA-1.5 demonstrate that PRN-Edit improves privacy protection while maintaining model utility.

**Strengths:**

1.	The paper introduces a precise method for mitigating privacy risks in LVLMs by targeting specific feature nodes, which allows fine-grained control without affecting overall model performance.
2.	The paper introduces a paired-sample dataset which provides a valuable resource for training and evaluating privacy-aware models, enhancing the generalizability of their approach.
3.	The paper conducts comprehensive experiments to validate their method, evaluating both on the proposed paired-sample dataset and on unseen datasets to test generalization. Additionally, they perform ablation studies for layer selection, demonstrating the effectiveness and contribution of each component in their approach.

**Weaknesses:**

1. The identification of privacy-risk nodes relies purely on gradient signals from cross-entropy losses rather than any causal or interpretable link to privacy-sensitive representations. Consequently, the learned mask may capture features correlated with output refusal behavior rather than truly encoding private information.
2.	Although the authors note that their two-module method is computationally less efficient than single-stage optimization in discussion, they do not provide any experimental data to support this claim. Without quantitative comparisons of runtime or resource consumption against baselines, the discussion on efficiency remains speculative and lacks empirical validation.
3.	Although PRN-Edit achieves higher refusal rates than baseline methods on unseen datasets (ScienceQA and MLLMGuard), the evaluation does not analyze the cases where the model still fails to prevent privacy leakage. It remains unclear whether these failures are due to insufficiently learned mask features, unseen privacy attributes during training, or other limitations? Therefore, the actual generalization capability may still be constrained by the diversity and coverage of privacy-related attributes in the training data, and this aspect warrants further discussion.
4.	The learnable mask is trained on a paired-sample dataset. The success of generalization likely depends on how well the privacy categories in training cover real-world sensitive information.
Is there a risk that the mask overfits to the specific question phrasing or token locations seen in training?
5.	The evaluation of model utility primarily focuses on response rates to benign questions, without assessing whether the model’s reasoning quality or task-specific accuracy is preserved after PRN-Edit. Adopting comprehensive utility metrics is necessary to validate that the method maintains overall performance, which is crucial for broader real-world applications, especially when considering efficiency concerns noted in Weakness 1.

**Questions:**

please check the weakness

---

### Note · Authors · 2025-11-12

I have read and agree with the venue's withdrawal policy on behalf of myself and my co-authors.